# Nosocomial Infections: Do Not Forget the Parasites!

**DOI:** 10.3390/pathogens10020238

**Published:** 2021-02-19

**Authors:** Ursula Fürnkranz, Julia Walochnik

**Affiliations:** Institute for Specific Prophylaxis and Tropical Medicine, Medical University of Vienna, Kinderspitalgasse 15, 1090 Vienna, Austria; julia.walochnik@meduniwien.ac.at

**Keywords:** nosocomial infections, parasites, immunosuppression, blood transfusion, transplantation, during birth, person-to-person contact, contaminated water/food

## Abstract

Nosocomial infections (NIs) pose an increasing threat to public health. The majority of NIs are bacterial, fungal, and viral infections; however, parasites also play a considerable role in NIs, particularly in our increasingly complex healthcare environment with a growing proportion of immunocompromised patients. Moreover, parasitic infections acquired via blood transfusion or organ transplantation are more likely to have severe or fatal disease outcomes compared with the normal route of infection. Many of these infections are preventable and most are treatable, but as the awareness for parasitic NIs is low, diagnosis and treatment are often delayed, resulting not only in higher health care costs but, importantly, also in prolonged courses of disease for the patients. For this article, we searched online databases and printed literature to give an overview of the causative agents of parasitic NIs, including the possible routes of infection and the diseases caused. Our review covers a broad spectrum of cases, ranging from widely known parasitic NIs, like blood transfusion malaria or water-borne cryptosporidiosis, to less well-known NIs, such as the transmission of *Strongyloides stercoralis* by solid organ transplantation or nosocomial myiasis. In addition, emerging NIs, such as babesiosis by blood transfusion or person-to-person transmitted scabies, are described.

## 1. Introduction

According to the European Center for Disease Prevention and Control (ECDC), 8.9 million healthcare-associated infections (HAIs) are estimated to occur every year in European hospitals and long-term care facilities [1]. Studies summarizing the results of 15 participating countries in Europe revealed that the most frequently isolated microorganisms from patients suffering from a nosocomial infection (NI) are bacteria such as *Pseudomonas aeruginosa*, coagulase-negative *Staphylococcus* spp., and *Escherichia coli* [1]. These results are quite similar to those found in other regions of the world, with the next most common microorganism being viruses [2]. However, parasites also contribute to HAIs. In a worldwide study conducted in 2007, it was shown, that parasitic infections account for 0.6 to 1% of infections acquired during a stay in hospital, depending on the geographic region (0.7% in Western Europe) [3]. These infections represent an increasing threat to patients and an increasing challenge to health care workers, especially in intensive care units (ICUs). In 2017, 11,788 (8.3%) patients staying in an ICU for more than two days faced at least one HAI [1]. NIs or HAIs mostly affect patients with impaired immunity and effectuate complications in addition to their original disease. Impaired immunity might be due to age (very young, very old), pre- and postsurgical status, disturbances in metabolism (e.g., diabetes), and of course, defects in immunity (innate, caused by immunosuppressive agents, or diseases, with AIDS being a prominent example).

The most important source of NIs is the patients themselves, who represent reservoirs for pathogens. Other sources can be health care personnel and visitors and even non-hospital personnel [4,5]. Moreover, food, water, blood transfusion, organ transplantation, and arthropods (themselves or as vectors) are further possible sources of NIs. 

This article gives an overview of parasites as causative agents of NIs, grouped by the main routes of infection. The main routes discussed are transfusion-mediated/solid organ transplantation (SOT)-mediated, during birth, contact with health care workers/other patients (person-to-person), water/food, and arthropods. Parasites, routes of infection, estimated frequency of infection, and selected references are summed up in Table 1. Moreover, short reference is given to the most common bacterial, fungal, and viral NIs in Table 2.

### 1.1. Blood Transfusion/Solid Organ Transplantation

In almost all parts of the world where transplantation is practiced, there is a major discrepancy between donor availability and the demand for organs. Thus, the justification for transplanting high-risk organs is based on the balance of risk in patients for whom delaying transplantation until a low-risk organ becomes available would increase this risk or even result in death [52]. However, it has to be taken into account that when parasites are present, a reactivation in the organ recipient of latent or dormant infections is a frequent event [53]. Here, we focus on infections due to infected organs/blood samples rather than on reactivated underlying infections. 

Theoretically, all parasites living in the human blood can be transmitted via blood transfusion and some also via solid organ transplantation (SOT). Thus, significant effort has been made to reduce pathogens in blood donor samples [54]. At the end of the 20th century, it was stated that “there is currently less than a one in a million chance that a blood transfusion within the United States will be complicated by a parasitic infection” [55]. However, in 2008, the Food and Drug Administration (FDA) announced that the protistan parasite *Babesia microti* was the most frequent transfusion-transmitted microbial pathogen in the United States, being associated with more than 100 cases of infection per year and the number of case reports increasing since then [13]. Nosocomial and blood transfusion-mediated infections with *Toxoplasma gondii*, *Plasmodium* spp., *Babesia* spp., *Trypanosoma cruzi*, *Leishmania* spp., and microfilariae are well-known. Moreover, emerging cases of SOT-mediated *T. cruzi* and *Strongyloides stercoralis* infections have been observed in the recent past. Rare cases of SOT-mediated infestations and infections with *Taenia solium*, *Schistosoma* spp., *Fasciola hepatica*, *Acanthamoeba* spp., and *Balamuthia mandrillaris* have been described. 

Infections with *Toxoplasma gondii* are the most common types of parasitic infection in humans (and many other warm-blooded animals) [56]. The infectious stages are tachyzoites (trans-placental transmission), bradyzoites (in meat), and oocysts (shed by cats and cat-like animals and persistent in the environment). *T. gondii* infections usually remain asymptomatic in immunocompetent individuals, whereas in immunocompromised patients, *T. gondii* can cause fatal encephalitis, myocarditis, pneumonitis, chorioretinitis, and generalized lymphadenopathy. SOT has been described as a source of *T. gondii* infections, especially in heart- and lung transplantations [6] but also in kidney transplantations [7]. SOT-related transmission of tachyzoites can only occur if the donor acquires the infection shortly before organ donation, whereas bradyzoites can be transmitted even if the donor is infected long before the transplantation. The incidence of SOT-transmitted toxoplasmosis is not precisely known, but it can reach 25–75% in the absence of prophylaxis. Moreover, reactivation of latent infections can occur due to organ transplantation and immunosuppression in general [6]. The worldwide prevalence of *T. gondii* oocysts in the feces of cats varies between 0% and 11% depending on the country investigated [8]. *T. gondii* oocysts have been described as being able to survive for up to 54 months in cold water [57]. Several outbreaks of waterborne toxoplasmosis have been described around the world [8]. Although nosocomial infections with *T. gondii* via contaminated water have not been described to date, this cannot be totally excluded, particularly in countries with many stray cats.

Between the years 1950 and 1972, 1756 cases of transfusion-transmitted malaria (TTM) were reported in 49 countries worldwide [58]. A study investigating TTM in non-endemic countries revealed 100 TTM case reports between 1911 and 2015, with marked differences between species: 45% involved *Plasmodium falciparum*, 30% *P. malariae*, 16% *P. vivax*, 4% *P. ovale*, 2% *P. knowlesi*, and 1% were mixed infections with *P. falciparum/P. malariae* [9]. The five *Plasmodium* species cause different forms of malaria, but all are naturally transmitted by the bite of a female *Anopheles* mosquito during its blood meal. *P. falciparum* malaria is an often fatal disease if left untreated. *P. ovale* and *P. vivax* malaria is characterized by febrile phases every 48 h, whereas *P. malariae* induces fever every 72 h, and *P. knowlesi* causes daily fever attacks. [59]. In 2018, an estimated 228 million cases of malaria occurred worldwide, resulting in 405,000 deaths, compared with 243 million cases and nearly 863,000 deaths in 2008 [60,61]. All *Plasmodium* species are able to survive in stored blood, even if frozen, and retain their viability for at least 1 week. After 14 days at 4 °C, *P. falciparum* has been shown to lose its ability to proliferate; however, parasites are still detectable after 28 days at 4 °C [62]. A significant issue of TTM is that malaria parasites are directly released into the recipient’s bloodstream (thus circumventing the development stages in the mosquito and the subsequent pre-erythrocytic phase in which the innate immune system of the host is normally activated), triggering the development of high-risk complications and potentially leading to a fatal outcome [63]. Malaria can also be transmitted via needle injuries [10,11], organ transplantation [12,64,65,66], as well as contaminated heparin locks [67,68,69], contaminated blood glucose meters [70], and even contaminated gloves [71]. 

Human babesiosis is a worldwide emerging disease caused by several species of *Babesia*—intra-erythrocytic protists transmitted primarily by hard-bodied ticks. Depending on the *Babesia* species involved, healthy individuals may remain asymptomatic and unaware of their infection, whereas immunocompromised patients are at increased risk for symptomatic disease. Symptoms most commonly include fever, chills, malaise, myalgia, gastrointestinal symptoms, and/or hemolytic anemia. Fulminant disease can manifest with disseminated intravascular coagulation, hemodynamic instability, and potentially fatal multi-organ dysfunction [72]. Babesiosis transmitted by transfusion has been reported throughout the world. *Babesia microti* has been rated the most common transfusion-transmitted pathogen in the United States, where from 1979 to 2009, 159 *B. microti* cases and three *B. duncani* cases due to blood transfusion were described [13]. Transfusion-mediated babesiosis has even found its way into the Centers for Disease Control and Prevention (CDC) graphs of the life cycle of *Babesia* spp., thus highlighting this important path of transmission [73]. In Europe, *B. divergens* is the most common species; however, no case of blood transfusion transmission of *B. divergens* has been documented to date [74]. Patients who experience transfusion-transmitted babesiosis are often seriously ill, and the mortality rate is about 20%. Moreover, 3.6% of transfusion-related deaths reported to the FDA from 2005 to 2010 were due to babesiosis [75]. In New York alone, 1.4% of all reported babesiosis cases over a period of 11 years were linked to transfusion [76]. Fang and McCullough [77] reported that despite a common mortality risk of 5–9% in hospitalized patients, this risk increased to 78% when the infection was acquired via transfusion. The fact that *Babesia* spp. reside within red blood cells prevents them from being cleared from whole blood through techniques such as leucocyte clearing. Moreover, many infected individuals, including potential blood donors, are asymptomatic and thus not aware of their infection, rendering their blood a potential risk for transmission [78]. It has been confirmed that, e.g., *B. divergens* is able to survive in stored blood bags at 4 °C for 31 days in sufficiently high numbers to lead to a high end-point parasitemia in blood recipients [79]. Therefore, and due to the rising numbers of cases of transfusion-mediated babesiosis, a protocol for the inactivation of *B. microti* in red blood cells was developed and tested in a hamster model, achieving not only pathogen reduction but also transmission inhibition [80]. 

*Trypanosoma cruzi*, the causative agent of Chagas disease, is naturally transmitted in the feces of hematophagous triatomines (“kissing bugs”) by unintentionally scratching it into the bite. Inside the human body, *T. cruzi* circulates for a short time extracellularly in the bloodstream and then becomes intracellular, invading cells from various tissues. A chronic infection can manifest for up to 30 years after infection and affect various internal organs (e.g., heart, esophagus, colon), resulting in dilatation of the respective tissue and, in severe cases, death [81]. Infections in immunocompromised patients show more severe progression and are more often fatal [82]. Several million people are infected worldwide, most of them in Latin America, and transmission of *T. cruzi* via blood transfusion has repeatedly been reported [14,83,84]. As of 2005, nearly 800 cases of infection with *T. cruzi* acquired by an iatrogenic route related to transfusion medicine practice had been collected since the first report of transfusion-mediated *T. cruzi* transmission in 1949 [15]. In addition, numerous cases of transmission of *T. cruzi* by SOT have been reported [80,85,86,87] and have to be distinguished from reports of *T. cruzi* reactivation after transplantation [88,89]. Most *T. cruzi* cases described occurred after transplantation of an infected heart or kidney, but liver and lung transplants have also been identified as carriers of *T. cruzi*. Huprikar et al. [87] detected *T. cruzi* in 9 out of 32 organ recipients of 14 *T. cruzi* positive donors. Six of the recipients died within one year after transplantation; however, death was only related to Chagas disease in one case. Thus, they concluded that although uninfected recipients who receive an organ from a *T. cruzi*-infected donor may develop acute Chagas, transmission via SOT is not universal. The shortage of available organs has already led to the usage of Hepatitis B-infected organs for transplant into vaccinated recipients with good outcomes [90]. Therefore, transplantation of organs from anti-*T. cruzi*-positive donors may be useful for patients who are in terminal stages of their illness. Benvenuti and colleagues [91] concluded that apart from the heart, other organs of *T. cruzi*-infected donors can be used for transplantation; however, caution must be taken.

The term leishmaniasis describes a series of diseases caused by different *Leishmania* spp. that are transmitted via the bite of a female sand fly. Within a human host, the flagellated promastigote forms develop into non-flagellated amastigotes that invade cells of the monocyte–macrophage lineage. Infections can be cutaneous, mucocutaneous, or visceral, and the severity of the disease depends on the *Leishmania* species as well as the immune state of the host. Visceral leishmaniasis, also known as “kala azar”, is accompanied by fever, swelling of the liver and spleen, and anemia, but often remains asymptomatic in immunocompetent individuals. The disease occurs in tropical and subtropical regions of all continents, except Australia, as well as in the Mediterranean basin, and is caused by members of the *L. donovani/infantum* complex [92]. Mucocutaneous leishmaniasis or “espundia” is most common in Bolivia, Brazil, and Peru, whereas cutaneous leishmaniasis (“oriental sore”) is distributed mostly in the Middle East but also in South America. The transmission of *Leishmania* spp. via blood transfusion is well known, but several endemic countries have addressed this issue, and due to improvements in blood component production, such as the development of leucocyte-reduction and pathogen-reduction technologies, the transmission risk is considered low today [18,93]. However, the use of these techniques is not a reality in many developing countries [94]. *Leishmania* amastigotes have been shown to survive and remain infectious after 30 days of storage under blood bank conditions [95]. *L. tropica* and *L. donovani* have been reported to survive in monocytes for 25 days in the red blood cell fraction when kept at 4 °C, for 30 days in unprocessed whole blood, and even longer (35 days) when the red blood cell fraction is conserved with additional glycerol. However, at 24 °C, these parasites survive for 5 days in the platelet fraction [96]. Numerous cases of visceral leishmaniasis have been described after kidney transplantations [97,98]; however, it is most likely that, in these patients, a latent, asymptomatic infection has become symptomatic because of the compromised immune state after transplantation, rather than the transmission of *Leishmania* occurring with organ transplantation. One case of suspected transmission of *L. infantum* via stem cell transplantation has been described in Switzerland [99]. As this patient presented with an unremarkable travel history, the infection was assumed to have been donor-acquired. 

Microfilariae is a general term for the larvae of roundworms of the Filarioidea type. They are transmitted by blood-sucking flies or mosquitos and are the causative agents of several different disease entities in humans. Lymphatic filariasis or elephantiasis can be caused by three different filarial species. While *Wuchereria bancrofti* in Asia is responsible for the highest number of infections worldwide, the disease can also be caused by *Brugia malayi* and *Brugia timori*. Lymphatic filariasis affects over 120 million people in 73 countries throughout the tropics and subtropics of Asia, Africa, the Western Pacific, and parts of the Caribbean and South America. The adult worms live in the human lymph vessels, resulting in dsfunctioning of the lymph system and, eventually, lymphedema. Impairment of the function of the lymph system makes it difficult for the body to fight other infections. Nevertheless, many people infected with *W. bancrofti* or *B. malayi*/*B. timori* have no symptoms and will never develop clinical symptoms [100]. Infections with *Mansonella* spp.—although widely present in Africa and equatorial America—have long been considered to be of minor medical relevance, as infected individuals often remain asymptomatic. However, more recent studies have shown that at least *M. perstans* may be responsible for a variety of clinical symptoms, including angioedema, pruritus, fever, headache, and pain in the bursae and/or joint synovia or in serous cavities [101]. Loaiasis, the so-called eye worm disease, is caused by *Loa loa* and is of significant medical relevance in West and Central Africa. Symptoms (if not asymptomatic) include itchy, non-painful swellings that come and go, mostly on the extremities of the body. The adult worms may occasionally crawl across the surface of the eye [102]. The only reported case of post-transfusional filariasis refers to *M. perstans* [19]. It has been assumed that, in endemic areas, transfused *M. perstans* microfilariae may be eliminated from the blood rather quickly, thus limiting the risk of post-transfusional disease [103]. However, allergic reactions in the recipients have been observed, arguing for the occurrence of transfusional transmission. Transfusion-transmitted microfilariae may be circulating in a recipient’s blood but do not develop into adult worms, and thus, allergic reactions due to breakdown products of dead microfilariae may manifest [104]. However, given that immunosuppressive therapy is not typically applied during transfusion, this does not represent an additional risk factor. In India, microfilariae have been identified in 10.6% of blood donors and 8.5% of recipients [104]. Studies from Nigeria, revealed the following rates of prevalence of microfilariae in blood donors: *L. loa* (1.3/3.5%), *M. perstans* (15.6%) and both parasites (0.2%) [105,106]. However, reports on *W. bancrofti* in blood donors are rare [107]. However, in these studies, the blood donors, rather than the donated blood, were screened; moreover, it cannot be excluded that the recipients already harbored microfilariae before transfusion. The presence of microfilariae in the peripheral blood is not constant over time. *W. bancrofti* is transmitted by nocturnal mosquitos; thus, the microfilariae mainly circulate in the peripheral blood during the night. The vector of *L. loa*, on the other hand, is the deer fly, which feeds during the daytime. For *M. perstans*, the periodicity has not been entirely elucidated, but microfilariae show a weak peak in the morning hours [108]. Blood collection is mostly performed during the day; thus, *W. bancrofti* is rarely found in blood samples. Donor blood from individuals living in endemic areas as well as from travelers to endemic areas should be screened for filarial parasites [109].

*Strongyloides stercoralis*, a common, tropical gastrointestinal helminth only causes minor pathologies in normal, healthy individuals, but it can be the cause of disseminated strongyloidiasis in immunocompromised individuals, particularly following transplant surgery, with larvae moving beyond the confines of the gut into other organs [110]. Infection is acquired when free-living infectious larvae penetrate the skin, migrate through the body and, eventually, reach the small intestine and develop into adults, which then mate and produce eggs. The first larvae hatch from the eggs in the intestine and are shed with the feces or can also autoinfect their host. Apart from natural infection, reactivation of a recipient’s chronic *S. stercoralis* infection following immunosuppressive therapy seems to be the most common mechanism of infection. Donor-derived *S. stercoralis* infection in SOT recipients is uncommon but is recognized as an emerging type of infection, resulting in a mortality rate of 34.6% in 27 reported SOT-associated *S. stercoralis* transmissions [20]. One impressive study referred to one single donor, whose heart, kidneys, and liver were transplanted and led to fatal outcomes in the heart and liver recipients; in the latter, however, *S. stercoralis* was not detected [111].

The uptake of larvae of *Taenia solium*, the pork tape worm, in undercooked meat leads to the development of an adult tape worm in the intestine without causing much harm to the human host. However, if eggs are ingested accidentally via fecally contaminated hands or food/water, larvae hatch in the intestine and infest various body tissues, eventually encysting as cysticerci. Neurocysticercosis is one of the most severe parasitic diseases, and it is the most common cause of epilepsy in the developing world. Three cases of transplant-associated neurocysticercosis have been reported in the literature, involving one kidney transplant recipient [21] and two recipients of liver transplants [22,112]. All patients recovered after treatment; however, for one of the liver transplant patients, it was stated, that he most likely already had neurocysticercosis before the transplant, and immune suppression after transplantation reactivated the formerly undetected infection [112].

*Schistosoma* spp. are the causative agents of intestinal (*S. mansoni* and *S. japonicum*) and urinary tract bilharziosis (*S. haematobium*). Transmission occurs when infested individuals contaminate freshwater sources with their excreta containing the eggs of respective parasites, which subsequently hatch in the water. The first larval stage (miracidium) develops in snails, which then release the infective larvae called cercariae. These actively penetrate the skin of the human host and develop during migration through various organs via the venous circulation. The adult worms settle in the mesenteric venules, the females residing within the males, but regularly depositing their eggs into the small venules of the portal and perivesical systems, respectively. Some of the eggs are shed with feces and urine, respectively, to continue the parasite’s lifecycle. Others become trapped in body tissues, causing granulomatous inflammation and progressive damage, as the eggs of all three species possess a spine, not only making them easy to identify under a microscope, but leading to damage to tissues and blood vessels. Schistosomiasis is prevalent in tropical and subtropical areas, particularly in poor communities without access to safe drinking water and adequate sanitation [113]. According to published data, the risk of *Schistosoma* transmission via liver transplants is low; therefore, it has been suggested that the presence of eggs in a liver biopsy, in the absence of any other liver complications, should not preclude organ donation, but that the recipient should be closely monitored for complications and treated, if needed. Data on kidney transplants remain more controversial [24].

Fascioliasis, the disease caused by the common liver fluke *Fasciola hepatica*, is a neglected tropical disease [114]. People usually become infested by eating raw watercress or other water plants contaminated with immature larvae of the parasite. These larvae penetrate the intestinal wall and migrate through the abdominal cavity and the liver tissue, finally reaching the bile ducts. Here, they develop into mature adult flukes that produce eggs, which are then released in the stools. One case of SOT-mediated fascioliasis with fatal consequences has been described in a child who underwent a liver transplant [25]. However, it was not possible to clarify whether the parasite indeed emerged from the transplanted liver or whether the child became infested after the transplantation and had a fatal course of disease due to immunosuppressive treatment.

Granulomatous amoebic encephalitis (GAE) is a rare disease caused by *Acanthamoeba* spp. (for a detailed description of this facultative pathogen, see below), *Balamuthia mandrillaris*, and possibly also other amoebozoan species found in the water and soil. The amoebae enter the body through the lower respiratory tract or skin lesions and may eventually migrate to the brain, causing GAE, which is usually fatal. Unlike *Acanthamoeba* spp., *B. mandrillaris* can also cause encephalitis in immunocompetent individuals. Two hundred human cases of GAE have been reported worldwide [115]; however, among these were several transplant-transmitted cases with incubation times of 17–24 days [26]. Most cases ended fatally for the transplant recipients [116,117]. In one series of transplant-transmitted *B. mandrillaris* infections, the donor’s brain was negative for *B. mandrillaris* by immunohistochemistry and PCR, while the transplanted organs were positive, and the donor serum revealed the presence of anti-*Balamuthia* antibodies [118]. Recently, *B. mandrillaris* was included into a risk score framework for the assessment of infectious encephalitis transmissible through SOT [119]. SOT-related GAE was also reported to be caused by *Acanthamoeba* spp. [120]. Of the 10 cases reported, only one patient survived. As in immunocompetent patients, GAE is usually fatal in SOT patients. However, in these patients, death usually occurs much faster than in immunocompetent patients, perhaps due to the profound impairment of cell-mediated immunity [120].

*Naegleria fowleri* is the causative agent of the so-called primary meningoencephalitis (PAME). *N. fowleri* is a widely distributed thermophilic ameboflagellate that grows at temperatures of up to 45 °C. Infection typically occurs during swimming, with the amoebae invading the brain via the olfactory nerve. The clinical course of PAME is rapid with a short incubation period of 3–5 days. The disease is characterized by severe frontal headache, fever, nausea, vomiting, and stiff neck, followed by seizures and coma, often ending fatally within ~1 week if not rapidly treated [121]. From 1962 to 2015, 138 cases have been reported in the USA [122]. Only a handful of patients have survived PAME [123]. The infection is assumed to be limited to the central nervous system (CNS). SOT of other organs from a PAME positive donor has been reported to not have provoked PAME in the recipients [124,125]. However, since 2009, the CDC have examined tissues from five patients with PAME and found evidence for extra-CNS dissemination of *N. fowleri* in four of the five cases. Thus, the potential risk of SOT-related transmission of *N. fowleri* must not be underestimated [28].

### 1.2. During Birth

Vaginal parasites, but theoretically also many intestinal parasites, can be transmitted to newborns. *Trichomonas vaginalis* is the most common non-viral sexually transmitted pathogen worldwide. It is a flagellated protist that causes symptomatic infections almost exclusively in women, leaving men infected but without symptoms. Typical symptoms include vaginitis and discharge; however, infection with *T. vaginalis* has also been linked to infertility [126] and cancer [127,128], as well as an enhanced risk for HIV acquisition and transmission [129,130]. The mode of *T. vaginalis* transmission to neonates has never been firmly established, but direct contact with the maternal genitourinary tract seems the most plausible explanation. *T. vaginalis* has been isolated from the urinary tract [131,132] as well as from the eyes [133], pharynx, and lower respiratory tract of newborns [29,134]. In most cases, only mild symptoms (vaginitis; respiratory problems) were observed and were cured with metronidazole. However, in one case, *T. vaginalis* was isolated from a brain abscess in a neonate, who was most probably infected during birth [135]. Premature rupture is rather frequently associated with maternal *T. vaginalis* infection during pregnancy [136], which likely also plays a role in the transmission of *T. vaginalis,* because prolonged exposure to the vaginal flora enhances the chance of neonatal colonization and infection [134]. One study performed in Spain revealed that despite an infection rate of 27.5% in the investigated mothers, no case of transmission to newborns was observed [137]. In another study, transmission was observed in 0.6% of neonates, although not all mothers were diagnosed with *T. vaginalis* [138]. In a very old study, 4.8% of neonates were found to be infected with *T. vaginalis* from their infected mothers [139]. Nevertheless, the real frequency of nosocomial infection by *T. vaginalis* remains unknown.

In addition, most intestinal parasites can be transferred to newborns during birth. This has repeatedly been reported for *Giardia* spp. [30], but also for, e.g., *Cryptosporidium* spp. [140] (for a description of the parasite, see the food/water section) and *Enterobius vermicularis* [37] (for a description of the parasite, see the person-to-person section). However, for many intestinal parasites, it is generally more likely that nosocomial infections occur via person-to-person contact (insufficient hand hygiene) or contaminated food/water, albeit detailed data are scarce. 

Finally, extracellular blood parasites (e.g., *Trypanosoma cruzi*) have also been described to be transmitted during birth, occurring in ~5% of children from infected mothers. These infections mostly remain silent in children but can be reactivated later in life [16].

### 1.3. Person-to-Person Contact

Hand hygiene has long been a challenge in hospitals and still is to some extent, although compliance of health care professionals has improved significantly [141,142]. Lacroix and Sørensen [38] described an outbreak of *Enterobius vermicularis* infections in a pediatric clinic with prevalence rates of between 22 and 29%. Enterobiasis is the most common helminthic infection, particularly in school-aged and pre-school children, with reported incidences varying, e.g., 3.6% in Slovakia, 11% in the USA, and 61% in India. After ingestion of eggs, the larvae hatch in the duodenum, develop into adults, and attach to the mucosa of the caecum and appendix. Female worms migrate to the perianal area to deposit their sticky eggs, resulting in itching, scratching, and, eventually, re-infection [143,144]. Humans are the only hosts of *E. vermicularis*, and person-to-person transmission (including autoinfection) via contaminated hands (eggs can remain infectious for several days, e.g., under the fingernails) is the most important method of transmission, although infection via contaminated objects (e.g., toys) or by inhalation (e.g., when making the beds) is possible [145]. 

A non-intestinal parasite transmitted via person-to-person contact is the itch mite *Sarcoptes scabiei*. Scabies is a widely distributed parasitic dermatosis that affects more than 130 million people worldwide at any point in time. *S. scabiei* mites burrow through the epidermis, their feces triggering a host immune response that leads to intense itching. Scabies is frequently complicated by bacterial secondary infections, which may result in septicemia and chronic nephritis [146]. In hospitals, immunocompromised patients or elderly, institutionalized patients admitted with unrecognized crusted (Norwegian) scabies constitute the main source for the spread of *S. scabiei*. At least 19 nosocomial outbreaks of scabies in 16 hospitals were identified by a literature review in 2006 [39]. Since then, numerous further reports of nosocomial scabies have been published, including also some large outbreaks, e.g., from Italy and the Netherlands [147,148,149,150]. 

### 1.4. Water/Food

As a general rule, drinking water in hospitals is not a risk for normal patients. However, for high-risk patients (immunocompromised, ICU-attenders), hospital tap water may become a source of nosocomial infections [151], as may poor hygiene practice in hospital kitchens [152]. Several parasite species may be transmitted to humans through the drinking water/food route, including, among others, the intestinal protists *Entamoeba histolytica*, *Giardia* spp., *Cryptosporidium* spp., *Cyclospora cayetanensis, Cystoisospora belli*, and *Balantioides (=Balantidium) coli*, and the generalized intracellular protists *Toxoplasma gondii* and *Trypanosoma cruzi*, as well as the intestinal helminths *Dibothriocephalus latum, Taenia* spp., and *Anisakid* nematodes, the liver flukes *Fasciola hepatica*, *Clonorchis sinensis,* and *Opisthorchis* spp., and *Trichinella* spp., which infect the human muscle tissue [23,153,154]. Moreover, the otherwise free-living amoebae *Naegleria fowleri* and *Acanthamoeba* spp. may infect humans after contact with water. Nosocomial infections have been reported for some of these water/foodborne (partly facultative) parasites. 

*Cryptosporidium* spp. are of medical and veterinary importance, being the causative agents of gastroenteritis in a variety of vertebrate hosts [155]. They need a host cell to replicate and after an asexual and a sexual cycle, robust oocysts are shed into the environment, where they are able to survive and remain infectious for months. While initially thought to only be associated with diseases in immunocompromised individuals, today, cryptosporidiosis is also considered a threat for immunocompetent humans [153]. Particularly, elderly patients with chronic illnesses represent a newly recognized category of persons at risk for *Cryptosporidium* infection, with many cases thought to be acquired institutionally [156]. In hospitalized children, *Cryptosporidium* infections occur most frequently in malnourished children, often leading to death [157]. Transmission from animals to humans has been described in veterinary clinics [158], but a more common route of *Cryptosporidium* transmission is via water/food [34,159] or via house flies [160], also in hospital environments. Direct person-to-person transmission of *Cryptosporidium* has also been described to occur; however, the reported frequency varies from uncommon [35,161] to rather often [33,36], including transmission via health care professionals [162]. Moreover, *Cryptosporidium* spp. can also survive on surfaces [33]. 

Infections with *Giardia* spp. account for around 280 million new cases worldwide per year [163]. The most common clinical signs of infection are abdominal pain with cramps and frequent watery diarrhea that later becomes bulky and greasy. Some patients experience a self-limiting mild illness, while others suffer from a long-lasting severe disease that does not respond to standard treatment [164]. Transmission can occur via contaminated food [165], person-to-person or animal contact, as well as contaminated water [166]. Transmission of *Giardia* spp. has even been reported from treated water supplies that meet water quality standards, as well as in hospital settings [34]. A characteristic of *Giardia* is the extremely low infectious dose (10 cysts or less) [167]. Nosocomial *Giardia* infections are not uncommon. For example, *Giardia* cysts and/or trophozoites were found in 4.4% of patients suffering from diarrhea in a hospital in Turkey [31] and 6.75% of patients in two hospitals in Shanghai [32]; moreover, *Giardia* was the most frequently identified and nosocomially acquired enteric pathogen in Jamaican children in a previous study [168]. 

*Acanthamoeba* spp. have been isolated from tap water, water treatment plants, air conditioning units, plumbing systems, drinking water networks, and cooling towers, as well as from shower heads in hospitals [169,170,171]. In addition to their potential to cause CNS infections (mainly in immunocompromised individuals), they may also cause an infection of the eye, so-called *Acanthamoeba* keratitis [172]. Disseminated acanthamoebiasis and granulomatous amoebic encephalitis (GAE), presumably acquired by contact with contaminated water, have been described in immunosuppressed patients [120,173]; moreover, one case of peritonitis that was a consequence of an *Acanthamoeba* infection through contaminated devices used for continuous ambulatory peritoneal dialysis has been reported [27]. Another issue concerning *Acanthamoeba* spp. is their potential to act as vectors for various bacteria (e.g., *Legionella pneumophila*, *Mycobacterium avium*), fungi (e.g., *Cryptococcus neoformans*), and viruses (e.g., Mimiviridae), many of them with clinical importance for humans [174]. Thus, the presence of free-living amoebae in hospital water systems represents a twofold issue: the amoebae being facultative pathogens themselves but also vectors for other pathogens. 

Infections with *Entamoeba histolytica* may remain asymptomatic but may also cause dysentery or extra-intestinal disease (amoebic liver abscess). The mode of infection is via oral uptake of the cysts from contaminated hands, water, or food. Following excystation in the intestine, *E. histolytica* trophozoites aggregate, divide, and form new cysts, thus effectuating self-limiting diarrhea. However, the trophozoites can also invade the colonic epithelium, causing colitis and eventually spreading to the liver and other organs, resulting in liver abscess and, very rarely, secondary amoebic encephalitis. Once a global burden, today, the occurrence of amebiasis is limited to countries with poor sanitary conditions [175]. A study from 1991 found *E. histolytica* to be the most commonly isolated pathogen in nosocomial diarrhea (besides yeasts) in a hospital in Mexico City [40]. Another study published in 2002 revealed that 19.5% of patients with dysenteric syndromes in a clinic in Dakar were positive for *E. histolytica*; thus, good management of patients and their excrement is required [41]. 

The possibility of transmission of *Trypanosoma cruzi* via food was established at the beginning of the 20th century. According to Anez and Crisante, 73% of the parasites remained alive for up to 72 h after contaminating various foods with feces of kissing bugs containing *T. cruzi* [176]. In some areas of Brazil, this oral route of transmission accounts for more cases of Chagas disease than the “classical” pathway via vectors [17].

### 1.5. Arthropods

The role of insect larvae and mites as infectious agents in nosocomial diseases seems to be rather small but constant [177]. The term myiasis refers to human and animal diseases caused by fly larvae (maggots), with larval development mainly occurring in necrotic tissue or at sites of excretion or secretion in anatomical cavities. Typical species are *Dermatobia hominis* (the human botfly), *Cordylobia anthropophaga* (the tumbu fly), *Cochliomyia hominivorax* (the new world screwworm fly), *Sarcophaga* spp. (the flesh flies), the green-bottle fly (*Lucilia* spp.), and also the common housefly (*Musca domestica*). While the latter two do not need a host for development but may develop in open wounds, several species depend on a host and can also invade intact skin. Several cases of most likely hospital-acquired myiasis, including cases in ICU patients, have been reported, e.g., involving *Lucilia sericata, Megaselia scalaris,* and *Parasarcophaga ruficornis* [178,179,180,181]. Nosocomial-acquired myiasis has been reported in intact skin, e.g., involving *Sarcophaga* species [42,182]. Although more common in the tropics, nosocomial myiasis may generally occur anywhere in the world, and infesting maggots are not only a medical issue themselves but may also carry other pathogens, superinfecting wounds caused by maggots [43]. Infestation by maggots is facilitated when a patient’s mental and/or physical functions are impaired, and they cannot brush away the flies. Similarly, small infants cannot defend themselves against approaching flies [177]. Moreover, flies can carry pathogens on their extremities, and thus represent a potential threat to immunocompromised and ICU patients [44]. Further, cockroaches (Blattodea) and the pharaoh ant (*Monomorium pharaonis*) have the potential to disseminate bacteria [45,46]. Cockroaches have a flattened body, so they can hide perfectly in cracks and small openings. Pharaoh ants were imported to Europe from tropic regions, and they thrive in buildings almost anywhere, even in temperate regions, provided central heating is present. Furthermore, pharaoh ants show aggressive behavior and may also bite patients [183]. 

Due to the large pigeon populations in many cities, *Dermanyssus gallinae,* the poultry red mite can be an issue in hospital settings. These mites normally suck blood from the breasts and legs of birds during the night but can also feed on humans, where infestation is known as gamasoidosis and presents as dermatitis and rash. Nosocomial infestations have been associated with, e.g., pigeon nests near hospital windows or with pigeons roosting in or on air conditioning systems [47,184].

Lice are obligate worldwide occurring parasites. They are 1 to 3 mm long and feed by sucking blood. Three varieties are parasitic to humans: *Pediculus humanus capitis* (head louse), *Pediculus humanus corporis* (body louse), and *Phtirus pubis* (pubic louse). *Pediculus humanus capitis* is a frequent community health concern infesting >100 million people worldwide annually, mostly children. Although head lice do not have a vector capacity for human pathogens, they can, of course, carry bacteria with them that can infect the sucking area secondarily. Moreover, the associated itching and embarrassment can be incriminatory to the patient. Adult lice and eggs (nits) can survive for up to 3 and 10 days, respectively, away from the host [185]. Head-to-head contact is the most important mode of transmission, followed by infested hats, hairbrushes, combs, towels, and bedding [186]. Treatment with shampoo containing permethrin, ivermectin, or benzyl-alcohol is recommended, as well as combing the hair with a special comb to get rid of nits [185]. In a pediatric clinic in India, 71.1% of female children and 28.8% of male children investigated within one year were infested with head lice [48]. Although the number of children who acquired their lice in the hospital was not investigated, the authors stated that the more people sharing a room, the more likely it is that the patients will be infested with lice therefore, head lice can become a nosocomial threat. *Pediculus humanus corporis* resides in the clothes instead of the hair and is mostly correlated with poor hygiene standards and/or homelessness. Infestation rates vary between 6 and 30% [187]. Body lice are known vectors of several rare but potentially life-threatening pathogens, like *Borrelia recurrentis* (causative agent of the louse-borne relapsing fever), *Rickettsia prowazekii* (typhus), and *Bartonella quintana* (trench fever) [187]. If a body-louse-infested patient is brought into a hospital setting, nosocomial spread of the lice and, thus, the bacterial diseases might occur [49]. This should be avoided by regular showering, changing/hot washing of clothes, and treating the respective bacterial infection. *Phtirus pubis* prefers pubic hair and is transmitted via close physical contact, including sexual contact and shared sleeping arrangements; however, clothing, bedding, and towels used by an infested person can act as vehicles for the lice. The incidence of pubic lice infestations is estimated to be between 1.3 and 4.6% worldwide and depends on the trend of shaving pubic hair, or not [50]. However, eyelashes and other types of body hair can also be infested with *Phtirus pubis* [188]. Although no cases of nosocomial infections with pubic lice have been published so far, low hygienic standards could lead to nosocomial infestations with pubic lice. *Demodex folliculorum* and *D. brevis* are normal inhabitants of the skin of many mammals, including humans. They live inside hair follicles in skin areas where sebum production is high. Newborns do not have *Demodex* mites; however, they acquire them shortly after birth, presumably via skin contact with their parents. Normally *Demodex* mites are present in low numbers (2–6 mites per follicle); however, immunosuppression has been shown to increase the number of *Demodex* sp., possibly leading to the clinical signs of blepharitis and conjunctivitis or rosacea, acne, and folliculitis (depending on the infection site) [189]. Moreover, a very recent study detected *Demodex* sp. in 56% of patients hospitalized for worsening heart failure in comparison to 25% in a healthy control group [51]. These data do not prove that infection with *Demodex* sp. was acquired in the hospital, but rather, that immunosuppression had an effect on these parasites, similar to reactivation in, e.g., *Leishmania* infections in kidney transplant patients (see above). 

## 2. Selected Most Common Bacterial, Fungal, and Viral NIs 

The majority of nosocomial infections are caused by bacteria, fungi, and viruses. Table 2 gives an overview of common nosocomial infections. 

In the most recent ECDC report on nosocomial bacterial infections [1], including data from 14 European countries, the most frequently observed disease was pneumonia, followed by blood stream infections and urinary tract infections. Pneumonia was associated with the use of intubates in 97.3%, and the most commonly isolated pathogens were *Pseudomonas aeruginosa*, followed by *Staphylococcus aureus*. *P. aeruginosa* is a Gram-negative, multidrug-resistant opportunistic pathogen that can be isolated from soil, water, skin flora, and most man-made environments throughout the world. This bacterium also has the potential to form biofilms, making it even harder to treat. *S. aureus* is a Gram-positive coccoid commensal bacterium of the microbiota of the human body; however, some strains of *S. aureus* are associated with severe infections in humans [190]. Bloodstream infections in HAI are associated with catheter usage in 37% of all cases. The most frequently isolated pathogens include coagulase-negative staphylococci and *Enterococcus* spp. [1]. *S. epidermidis* and *S. saprophyticus*, coagulase-negative *Staphylococcus* species, are commensals of the skin and/or urinary tract but can cause severe infections in immunosuppressed patients. Enterococci are Gram-positive cocci and normally intestinal commensals, even belonging to the obligate gut flora. However, they can cause life-threatening infections in immunocompromised persons. Urinary tract infections in ICUs are mostly associated with the use of urinary catheters (97.9%), and the most frequently isolated pathogens are *Escherichia coli* and species of the already mentioned genus *Enterococcus*. *E. coli* also belongs to the obligate intestinal flora; however, in the wrong habitat (e.g., the urinary tract), it may cause severe infections [191]. Hand hygiene is of major importance for the prevention of nosocomial infections, particularly in settings with immunosuppressed patients. Involved are, most frequently, Gram-negative bacteria and *Bacillus* spp. [192]; however, anaerobic bacteria, such as *Fusobacterium* sp. and *Clostridium* spp., are also isolated from nosocomial infection-sites of cancer patients [193]. As all the described bacteria can be found in/on healthy human bodies, the most likely source of infection and contamination of catheters/tubi is either the patient himself/herself or health care personnel.

Members of the genus *Aspergillus* can cause a variety of diseases, subsumed as Aspergillosis. Aspergillosis is assumed to affect more than 14 million people worldwide. Invasive aspergillosis occurs infrequently in SOT recipients but with a high mortality rate of 40% [194]. However, it is not entirely clear as to whether these infections are due to a (re-)activation of already present *Aspergillus* spp. or whether they are acquired during transplantation through contaminated air, ventilation systems, or air filters, or if they are derived from the organ itself if the donor is infected [195]. Certain *Candida* species, especially *Candida albicans*, are part of the human microbial flora, but in critically ill patients, they are a source of candidemia. Between 2009 and 2010, *Candida* spp. were the fifth most common pathogen identified in HAIs reported to the CDC. Nosocomial fungal infections are most likely due to intravascular catheters colonized by *Candida* spp. from the patient’s endogenous microflora or *Candida* spp. acquired from the healthcare environment [196].

Viruses account for 1–5% of nosocomial infections [197]. Nosocomial spread of viruses often parallels outbreaks in the community. Many viruses are spread via aerosols, and thus visiting relatives and health care personnel are the most important sources of infection. Respiratory viruses (e.g., respiratory syncytial virus (RSV), influenza viruses, rhinoviruses, coronaviruses, and adenoviruses) are increasingly being recognized as significant pathogens associated with seasonal nosocomial outbreaks. RSV is a major cause of morbidity in infants and young children; influenza mostly commonly affects elderly persons and has an associated mortality rate of 50%. Rhinoviruses and coronaviruses (exclusive of SARS-CoV-2) are responsible for up to 40% and 15% of cases of the common cold, respectively. Other aerosol-spread viruses, like measles, mumps, and rubella viruses, do not play huge roles in nosocomial infections anymore, thanks to vaccination programs. Neither rubella nor mumps infections have more severe etiopathologies in immunocompromised individuals, but such patients can develop severe progressive measles infection associated with giant cell pneumonia, with an associated mortality rate of 70%. Gastrointestinal viruses can be spread via the feco–oral route, the most prominent example being rotaviruses. Rotaviruses have been identified as the cause of diarrhea outbreaks in the elderly as well as in children, of whom up to 70% shed the virus in their stools. Blood-borne viruses, such as hepatitis B virus, hepatitis C virus, and human immunodeficiency virus type 1 (HIV-1) have all been associated with nosocomial infections. While the risk of an infection with hepatitis B upon percutaneous contact with an infected individual is up to 30% (in unvaccinated persons), the average risk of transmission of HIV was calculated to be 0.32% [198].

All infection scenarios described involved highly immunocompromised individuals in intensive care units, cancer units, or pediatric clinics. However, one has to keep in mind that many of the described bacterial, fungal, and viral nosocomial infections can also occur in e.g., outpatient clinics [199] or dental clinics [200].

**Table 2 pathogens-10-00238-t002:** Common agents of nosocomial infections (bacteria, fungi, viruses).

Pathogen	Source of Infection	Estimated Frequency	Selected References
**Bacteria**			
*Pseudomonas* *aeruginosa*	intubation	7.2–33.3%	[1]
*Staphylococcus aureus*		3.3–30.6%	
Coagulase negative Staphylococci	blood catheter	9.5–45%	
*Enterococcus* spp.		7.9–53%	
*Escherichia coli*	urinary catheter	14–44%	
*Enterococcus* spp.		9–37.5%	
**Fungi**			
*Aspergillus* spp.	SOT	23 cases	[194]
*Candida* spp.	surgical infection/catheter associated	10%	[196]
respiratory viruses	aerosols	40% in winter season	[198]
mumps, measles, rubella	aerosols	rare	
gastrointestinal viruses	feco–oral	45–50%	
blood-borne viruses	body fluids, needles	Up to 20%	

## 3. Search Strategy and Selection Criteria

This review was conducted using PubMed, as well as text books available in the library of our working group (Molecular parasitology at the Institute for specific prophylaxis and tropical medicine; Medical University of Vienna). PubMed was searched by entering the respective names of the parasites and “nosocomial” or “hospital acquired”, “transplantation”, “food/water”, “person to person”, “during birth/congenital”. No date limit was set; also, articles published in the first half of the twentieth century were included. Moreover, articles published in English, French, Spanish, and German were included. Reports dealing with, e.g., transmission via water/food and not hospital settings and reports only describing the reactivation of parasitic diseases instead of transmission due to SOT/blood were excluded. However, reports allowing a comparison of different routes of transmission were included. To allow a comparison with bacterial, fungal, and viral NIs, the words “nosocomial”, “transplantation”, or “immunosuppressed” were entered into PubMed. 

## 4. Conclusions

Compared with bacterial, fungal, and viral NIs, the prevalence of nosocomially acquired parasitic infections is low, and thus awareness among health care personnel is also usually low, particularly in non-endemic countries. As a result, diagnosis and treatment may be significantly delayed. However, several parasitic infections are considered to be emerging, e.g., transfusion-transmitted babesiosis or nosocomially acquired cryptosporidiosis in elderly patients with chronic illnesses. As the number of immunocompromised patients is rising constantly, and transfusion and transplant medicines are becoming more widely available, parasites have to be considered when confronted with complications in the ICU. Moreover, the roles of some parasites in nosocomial infections have only been recognized rather recently, e.g., the role of amoebae as vehicles for legionellae and other bacteria in hospital water and air conditioning systems. Clearly, nosocomial parasitic infections are not restricted to tropical regions, and parasites should be included in infection control and intervention strategies in hospital settings and organ transplant screening worldwide.

## Figures and Tables

**Table 1 pathogens-10-00238-t001:** Nosocomial parasites, listed according to their route of infection, with their infective stages, routes of infection, reported frequencies, and selected references.

Pathogen	Infective Stage in Healthcare-Associated Infection (HAI)	Route of Infection in HAI	Reported Frequency	Selected References
*Toxoplasma gondii*	bradyzoites (infection of donor a long time ago) tachyzoites (recent infection of donor)	solid organ transplantation (SOT; heart, lungs, kidneys)	25–75% in absence of prophylaxis	[6,7]
	oocysts	water	unknown	[8]
*Plasmodium* spp.	schizonts of the erythrocytic cycle	blood transfusion	1 case/year in non-endemic countries	[9]
		infected needles	very rare	[10,11]
		SOT	unusual	[12]
*Babesia* spp.	trophozoites	blood transfusion	162 cases in 30 years	[13]
*Trypanosoma cruzi*	metacyclic trypomastigotes	blood transfusion	800 cases in 2005	[14,15]
		during birth	5% of children of infected mothers	[16]
		contaminated food/water	in Brazil: more infections than classical route (via kissing bugs)	[17]
*Leishamania* spp.	amastigotes	blood transfusion	~6% of blood samples positive for *L. infantum* DNA	[18]
Filariae	microfilariae	blood transfusion	very rare	[19]
*Strongyloides stercoralis*	larvae	SOT	uncommon; 27 reported cases	[20]
*Taenia solium*	cysticerci	SOT	3 cases	[21,22]
	eggs	food/water person-to-person	unknown	[23]
*Schistosoma* spp.	eggs	SOT	very few cases	[24]
*Fasciola hepatica*	adult flukes	SOT	1 case	[25]
*Acanthamoeba* spp./ *Balamuthia mandrillaris*	trophozoites	SOT	>10 cases	[26]
		contaminated devices	1 case	[27]
*Naegleria fowleri*	trophozoites	SOT	not reported to date	[28]
*Trichomonas vaginalis*	trophozoites	during birth	2 to 17% of neonates of infected mothers	[29]
*Giardia* spp.	trophozoites and cysts	during birth	rare	[30]
		food/water	4.4–6.75% of diarrhea patients	[31,32]
*Cryptosporidium* spp.	oocysts	during birth	rare	[33]
		food/ water	often	[34]
		person-to-person	uncommon–rather often	[35,36]
*Enterobius vermicularis*	eggs	during birth	rare	[37]
		person-to-person	20–30% in pediatric clinics	[38]
*Sarcoptes scabiei*	mainly impregnated female mites	person-to-person	19 outbreaks in 16 hospitals	[39]
*Entamoeba histolytica*	cysts	food/water	probably common in clinics with poor sanitation	[40,41]
Maggots	larvae	female flies lay eggs in open wounds/intact skin	low frequency, but constant	[42]
Flies	bacteria/viruses	bacteria/viruses mechanically transported	e.g., 42% positive for *Escherichia coli*, 96% positive for *Pseudomonas* spp.	[43,44]
Cockroaches	bacteria/viruses	bacteria/viruses mechanically transported	detected in 70% (German cockroach) and 40% (Oriental cockroach) of hospitals in Poland	[45]
Pharaoh ants	bacteria/viruses	bacteria/viruses mechanically transported	detected in 14% of hospitals in Poland	[46]
*Dermanyssus gallinae*	mites	bird nests near window provide mites that feed on blood	rare	[47]
*Pediculus humanus capitis*	adults/nits	close hair contact	30–70% in pediatric clinics	[48]
*Pediculus humanus corporis*		contact with infested clothing	unknown; (6–30% in homeless people)	[49]
*Phtirus pubis*	adults	sexual contact; contact with bedding	unknown; (general population: 1.3–4.6%)	[50]
*Demodex folliculorum/Demodex brevis*	adults	immunosuppression increases number of parasites	56% in heart failure patients	[51]

## Data Availability

Not applicable.

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
