# Peer review of "Nosocomial Infections: Do Not Forget the Parasites!"

_pathogens, 2021, doi:10.3390/pathogens10020238_

Round 1

Reviewer 1 Report

The article forms a very good study on hospital infections caused by parasites. The scope of the study is considerable as both the frequency of infections and various ways of their transmission are taken into account.

However, the study seems to lack information about hospital infections caused by head lice (Pediculus humanus capitis), clothes lice (Pediculus humanus corporis) and pubic lice (Pediculus pubis). Moreover, it could be worthwhile to mention infections caused by mange mites (Demodex folliculorum) or Demodex mites (Demodex brevis). 

The lack of any information about these parasites in the reviewed article might be seen as a form of research negligence especially because lice might be spread by direct contact and they might be transmitters of typhus or trench fever which should be mentioned in the survey.  

Author Response

Thank you for your very helpful suggestions. We have carefully revised our manuscript taking into account all comments made. Below we have responded to all comments, point by point. We are sending along a corrected version of the revised manuscript with all changes made highlighted in yellow.

Moreover, we have had our manuscript corrected by the MDPI English editing service, the certificate is attached.

Reviewers’ comments

Reviewer 1

The article forms a very good study on hospital infections caused by parasites. The scope of the study is considerable as both the frequency of infections and various ways of their transmission are taken into account.

However, the study seems to lack information about hospital infections caused by head lice (Pediculus humanus capitis), clothes lice (Pediculus humanus corporis) and pubic lice (Pediculus pubis). Moreover, it could be worthwhile to mention infections caused by mange mites (Demodex folliculorum) or Demodex mites (Demodex brevis).

The lack of any information about these parasites in the reviewed article might be seen as a form of research negligence especially because lice might be spread by direct contact and they might be transmitters of typhus or trench fever which should be mentioned in the survey. 

Thank you for this helpful comment. We totally agree and now included head and body lice, as well as pubic lice and Demodex spp. to our study (p.11-12, Tab.1).

Reviewer 2 Report

The manuscript covers nosocomial infections of parasites and does have some important information. The title gives the impression that the authors speak generally about nosocomial infections with a specific focus on parasites. This is not the case, as the authors exclusively focused on parasites. I suggest that the authors attempt to compare some of their findings with parasites to bacterial nosocomial infections from different clinical and environmental settings. For example, the authors should indicate the frequency of parasitic infections in cancer patients receiving treatment in cancer centers and cancer hospitals. They can compare this to findings with bacterial nosocomial infections in similar settings (DOI: 10.2217/fmb.12.125). Similarly, the authors should indicate the frequency of parasitic infections acquired from dental clinics and compare this to findings with bacterial nosocomial infections in similar settings (DOI: 10.1038/s41598-017-07713-8). The authors should do the same thing with other clinical and environmental settings. There should be a separate section dedicated to this comparison. Otherwise, the title will not be reflective of the content. There should also be a table that summarizes this new section. Currently, there is only one table which is not enough.

Minor issue: The abstract has a formatting issue that needs to be fixed. 

Author Response

Thank you for your very helpful suggestions. We have carefully revised our manuscript taking into account all comments made. Below we have responded to all comments, point by point. We are sending along a corrected version of the revised manuscript with all changes made highlighted in yellow.

Moreover, we have had our manuscript corrected by the MDPI English editing service, the certificate is attached.

Reviewer 2

The manuscript covers nosocomial infections of parasites and does have some important information. The title gives the impression that the authors speak generally about nosocomial infections with a specific focus on parasites. This is not the case, as the authors exclusively focused on parasites. I suggest that the authors attempt to compare some of their findings with parasites to bacterial nosocomial infections from different clinical and environmental settings. For example, the authors should indicate the frequency of parasitic infections in cancer patients receiving treatment in cancer centers and cancer hospitals. They can compare this to findings with bacterial nosocomial infections in similar settings (DOI: 10.2217/fmb.12.125). Similarly, the authors should indicate the frequency of parasitic infections acquired from dental clinics and compare this to findings with bacterial nosocomial infections in similar settings (DOI: 10.1038/s41598-017-07713-8). The authors should do the same thing with other clinical and environmental settings. There should be a separate section dedicated to this comparison. Otherwise, the title will not be reflective of the content. There should also be a table that summarizes this new section. Currently, there is only one table which is not enough.

Thank you very much for your helpful comments and suggestions.

We have now incorporated a new section discussing most common bacterial, fungal, and viral nosocomial infections (p.12-14). We did not go too much into detail as this would go beyond the scope of this review, which is focussed on parasites. However, we agree that the comparison with other nosocomial pathogens is highly interesting and also important; therefore we also added a new table comparing the (estimated) frequencies common bacterial, fungal and viral nosocomial agents (p.14). Also, we added more details and different patient cohorts, now including also cancer patients and patients in outpatient centres as dental clinics.

Thank you also for the interesting literature, which we now also included into our paper.

Moreover, in the conclusion section (p. 14-15), we now also referred to the new section and the higher rates of nosocomial bacterial, fungal and viral infections in intensive care units, highlighting that – as these infections are much more common and more known by healthcare personnel – it is even more important not to ignore infections that are rare like parasitic ones, or thought only to occur in tropical regions.

Minor issue: The abstract has a formatting issue that needs to be fixed.

Sorry for this mistake, this has been fixed.

Round 2

Reviewer 2 Report

NA